# An attention-based recurrent learning model for short-term travel time prediction

**Jawad-ur-Rehman Chughtai** [1,2]*, **Irfan Ul Haq**[1,2], **Muhammad Muneeb**[3]

**1** Department of Computer and Information Sciences (DCIS), PIEAS, Islamabad, Pakistan, **2** Digital Disruption Lab, DCIS, PIEAS, Islamabad, Pakistan, **3** Department of Mathematics, Khalifa University of Science and Technology, Abu Dhabi, United Arab Emirates

* jawadchughtai@gmail.com

**Data Availability Statement:** All the implementation details of our work can be found at https://github.com/jawadchughtai/Att_GRU_TTP We have made the visibility of the repository

## Abstract

With the advent of Big Data technology and the Internet of Things, Intelligent Transportation Systems (ITS) have become inevitable for future transportation networks. Travel time prediction (TTP) is an essential part of ITS and plays a pivotal role in congestion avoidance and route planning. The novel data sources such as smartphones and in-vehicle navigation applications allow traffic conditions in smart cities to be analyzed and forecast more reliably than ever. Such a massive amount of geospatial data provides a rich source of information for TTP. Gated Recurrent Unit (GRU) has been successfully applied to traffic prediction problems due to its ability to handle long-term traffic sequences. However, the existing GRU does not consider the relationship between various historical travel time positions in the sequences for traffic prediction. We propose an attention-based GRU model for short-term travel time prediction to cope with this problem enabling GRU to learn the relevant context in historical travel time sequences and update the weights of hidden states accordingly. We evaluated the proposed model using FCD data from Beijing. To demonstrate the generalization of our proposed model, we performed a robustness analysis by adding noise obeying Gaussian distribution. The experimental results on test data indicated that our proposed model performed better than the existing deep learning time-series models in terms of Root Mean Square Error (RMSE), Mean Absolute Error (MAE), Mean Absolute Percentage Error (MAPE), and Coefficient of Determination ($R^2$).

## Introduction

Recent years have witnessed a drastic movement of people from rural to urban areas. 4.1 billion people were living in urban areas in 2017, comprising 55% of the total global population [1]. According to the Population Reference Bureau report, the current population will have grown by 14% by 2050 [2]. Urbanization has significantly improved the quality of life of individuals [3] whereas, on the other hand, it has brought new challenges and raised new concerns [4].

Advancement in information and communication technology has brought about a significant rise in the availability of mobility data collected through multiple data sources, including

public. Also, minimal anonymized dataset is provided in the Dataset folder to replicate our work.

**Funding:** The author(s) received no specific funding for this work.

**Competing interests:** The authors have declared that no competing interests exist.

FCD (Floating Car Data), detectors, cameras, etc. Research groups and companies analyze this data using big data and machine learning to improve people's living standards [5]. Researchers have used data from multiple sources to improve traffic-related operations with applications in traffic congestion prediction [6], traffic flow prediction [7], traffic speed estimation [8], traffic demand prediction [9], traffic signal control [10], parking space forecasting [11], stay point detection [12], traffic accident prediction [13], accident severity analysis [14], and many others.

One of the essential components of an Intelligent Transportation System (ITS) is Travel Time Prediction (TTP). Accurate TTP helps commuters and travelers make wise decisions about departure time and route selection which, in turn, leads to congestion avoidance. Moreover, it assists logistic operators in improving service quality and reducing transportation costs by avoiding congested routes. Furthermore, TTP helps traffic managers and decision-makers make traffic-related strategies and improve existing operations [15].

Various approaches including statistical (e.g., Historical Average (HA)), classical time-series (e.g., Auto-Regressive Moving Average (ARIMA) and variants), machine learning(e.g., Random Forest (RF), Support Vector Regression (SVR)), and deep learning-based approaches (Multi-Layer Perceptron (MLP), Convolutional Neural Network (CNN), Recurrent Neural Network (RNN), and variants) have been proposed to predict travel time [16–23]. Deep learning-based approaches outperformed their counterparts in prediction tasks because of their ability to deal with non-linearities, traffic trends, and long-term sequences [24].

Recurrent Neural Networks (RNNs) are specialized models developed for sequence learning problems. Simple RNN performs well for short-term sequences. However, it suffers from exploding Gradient and vanishing gradient problems when dealing with long-term sequences. Gated Recurrent Unit (GRU) and Long-Short-Term Memory (LSTM) were developed to resolve these issues. Both models have shown state-of-the-art performance on various sequence learning tasks with applications ranging from Natural Language Processing (NLP) to traffic prediction [25].

Existing RNNs architectures like LSTM and GRU suffer from implicitly modeling the context in historical travel time sequences. These models give equal weights to all hidden states when used for the TTP task. We introduced an attention mechanism, which aims to re-weight the network weights by leveraging the hidden relationship between distinct positions in the Travel Time (TT) sequence [26].

In this paper, we propose an attention-based GRU model for TTP. We selected GRU due to its simplistic architecture and faster training time. The experimental results on the Q-Traffic dataset show significant improvement in short-term traffic prediction compared to baseline approaches.

The main contribution of the paper can be summarized as follows:

- This paper proposes a deep learning model based on GRU for short-term travel time prediction. We introduced self-attention in GRU to address the limitation of GRU in finding the relation across various travel time positions in the input (past) sequences. To the best of our knowledge, no attempt has been made to forecast travel time using traffic flow as input with attention-based GRU.

- We compared our proposed model with baseline state-of-the-art statistical, classical time-series, Machine Learning (ML), and Deep Learning (DL) approaches. The comparative results on the Beijing-based FCD dataset show considerable improvement in Root Mean Square Error (RMSE), Mean Absolute Error (MAE), Mean Absolute Percentage Error (MAPE), and Coefficient of Determination ($R^2$).

- Moreover, we performed perturbation analysis by adding noise to our data which validated the generalization abilities of our proposed model.

We organized the remainder of this paper as follows: Section II provides the historical background of our studied area. Section III explains the proposed methodology. Section IV presents the findings and results of our work. Section V concludes the paper.

## Related work

The literature on TTP is grouped into two broad categories: traditional approaches and advanced approaches. Traditional approaches include classical approaches and machine learning-based approaches while advanced approaches include deep learning-based approaches, ensemble learning-based approaches and attention-based approaches.

### Traditional approaches for TTP

**Classical approaches.** Earlier travel time prediction approaches employed statistical theory-based modeling and classical time-series approaches. The HA was one of the first statistical theory-based modeling approaches used in TTP studies. In this approach, travel time in the historical period is averaged to get the prediction [16]. HA is computationally fast and doesn't require any assumption for prediction. However, HA does not consider temporal variations and features, resulting in lower prediction precision. The ARIMA was another widely used classical time-series model [17]. ARIMA treats the traffic data as a stationary time series to predict future travel time, and this assumption hampers ARIMA's ability to predict TT in uncertain or changing traffic conditions. Despite ARMIA's widespread use, the simple linear model falls short of accurately forecasting nonlinear traffic data.

**Machine learning-based approaches.** SVR is one of the widely used approach for TTP. Some studies used SVR with nonlinear transformations to handle data complexities. The authors in [18] proposed SVR for freeway TTP. When compared to classical Support Vector Machine (SVM), an SVM model optimized using the artificial fish swarm approach [27] or least squared loss function and equality constraint [28] has been found to improve model precision. k-Nearest Neighbors (k-NN), an example-based or pattern matching-based model, has also been widely employed for similarity pattern matching in travel time problems on urban roads and highways [29]. Typically, k-NN uses euclidean distance to find k similar patterns and then uses a weighted algorithm to get the final result. Myung et al. in [30] employed k-NN on data collected through automatic toll collection and vehicle detector systems to predict highway travel time.

### Advanced approaches

**Deep learning-based approaches.** DL has attracted researchers' attention for TTP and is still an enduring area. Different DL approaches, including MLP, auto-encoders, CNN, and RNN, have been applied for TTP. MLP is one of the earliest and most widely used approaches for TTP. The authors in [21] have proposed a multi-step deep learning approach for TTP. Extensive feature engineering is performed using geospatial feature analysis, principal component analysis, and k-means clustering, followed by a deep-stacked auto-encoder. The findings revealed that the proposed approach performed well for general traffic dynamics but failed to predict travel time in case of rare events. Fu et al. [31] used MLP as a final predictor on top of wide deep recurrent modules to predict travel time. Yuan et al. [32] employed MLP for spatio-temporal learning of travel time by exploiting periodicity in daily and weekly patterns and the road network structure. Researchers have also used CNN to capture the spatial aspects of TTP

data. The authors in [20] proposed a global-level representation for CNN to capture better the relationship between the predicted information and historical data points to overcome local receptive field limitations. However, the proposed approach is validated only on a single highway link. A new local-receptive field is proposed by [33] to model nonlinear spatiotemporal relationships in travel time data over multiple highway links. Shen et al. [34] implemented CNN with RNN to learn both spatial and temporal features to improve the prediction of FCD. Likewise, the authors in [35] proposed a graph convolutional network with LSTM to capture spatiotemporal features in urban road travel time data. A graph-based deep learning approach is presented in [36]. The model gives promising results compared to baselines, but only short trajectories are considered with no external features incorporated. Unlike MLPs and CNNs which are feed-forward neural networks and take data all at once, RNNs act on data sequentially and are frequently employed in the NLP domain. RNNs have also been widely adopted for TTP. Zhao et al. [37] employed GRU on integrated data from remote transportation microwave sensors and dedicated short-range communications to predict travel time. The experiment used two freeway segments yielding better results with data fusion. With the introduction of data sparsity as the spatial scale increased, a neighboring-segments-based strategy is proposed in [23] which employed GRU to predict travel time for the entire trajectory path. Adjacent road segment information addresses trajectory data sparseness due to longer trips. In [22], the authors compared the LSTM-based RNN model with a Back Propagation Neural Network (BPNN) and Deep Belief Network (DBN) using multi-factor data for TTP.

**Ensemble learning-based approaches.** Researchers have also employed ensemble approaches for TTP. The most extensively used ensemble approaches for TTP are Gradient Boosting Machine (GBM), eXtreme Gradient Boosting (XGBoost), and RF. The authors in [19] implemented GBM, RF, and ARIMA for multi-step ahead prediction using freeways data and demonstrated better performance of GBM over RF and ARIMA. Verkehr In Städten-SIMulationsmodell (VISSIM) freeway data is used in [38] to predict TT for multi-step ahead prediction. The Gradient Boosting Regression Tree (GBDT) model performed better than the SVM and BPNN. Chen et al. [39] proposed the XGBoost model to predict freeway TT. The results show better performance of XGBoost compared to GBM. The authors in [40] implemented decision tree, RF, XGBoost, and LSTM and demonstrated better performance of RF on freeway data. However, as the data volume increases, the performance of these approaches begins to deteriorate, and to improve the performance of these approaches, some studies used a combination of various algorithms for TTP. Ting et al. in [41] proposed an ensemble based on XGBoost and GRU to improve prediction accuracy using freeway data. For example, the results from the ensemble of light gradient boosting machine and MLP are combined using a decision tree for TTP in [42]. Recently, an ML-based ensemble has been proposed in [43] that uses kalman filter, k-NN, and BPNN as base learners. The prediction results are combined using fuzzy soft set theory to improve the prediction accuracy using freeway data. Although the proposed study improves individual models' prediction accuracy, the criteria for choosing these base learners are not discussed.

**Attention-based approaches.** Attention mechanisms or attention-based models have proven to be very powerful and adaptable models in a wide range of transportation applications [6, 8, 44]. Attention mechanism has recently been introduced in TTP due to its success in other related applications to learn only the relevant context. [26, 45] implemented an attention mechanism with LSTM to enhance performance. Ran et al. [46] introduced an attention mechanism with CNN on freeways data for better results. The authors in [47] employed attention with LSTM for joint prediction of travel time and next location.

To summarize, various approaches have been developed for TTP to improve and enhance TTP performance. Different datasets have been used in different studies. Some studies are

conducted on freeway data, while others use urban road networks. It is difficult to conclude which approach is better in every scenario. Generally, deep learning performed better in handling data complexities and non-linearities. For traffic data, RNNs have shown promising results due to the nature of the problem (i.e., the traffic conditions at the current timestamp or near future timestamp are dependent on past timestamps).

Therefore, we propose a GRU variant to improve prediction performance by adding an attention method that allows the model to learn only the relevant context rather than considering all historical timestamps (positions) equally. The attention mechanism resulted in enhanced feature space yielding better results.

## Proposed methodology

### Problem definition

Travel time prediction can be formalized as forecasting future travel time given historical travel time. Let $T_\tau^i$ denote the $i$-th segment travel time during $t$-th time period. Given the historical travel time sequence $T_\tau^i$ ($\tau$=t-m$\delta$,...,t-$\delta$, t and $i \in S$, where $S$ is the set of segments in the considered study area), the task is to predict segment travel time at time interval (t + f$\delta$) for some prediction horizon $\delta$. In this work, we consider $\delta$ = 15 minutes, $m$ = 4, and $f$ = 1, 2, 3, 4, which means that previous one-hour observations are used to predict the travel time of the next 15 minutes, 30 minutes, 45 minutes and 60 minutes.

### GRU

RNNs are proposed to process sequential data efficiently. However, standard RNNs suffer from exploding and vanishing gradient problems as the input sequence lengthens. To overcome these limitations, two specialized variants of RNNs, GRU, [48] and LSTM, [49] are developed. These models use the gated mechanism to handle long-term time-series sequences. LSTM comprises three gates; input, forget, and output gate. GRU uses two gates (update gate and reset gate), speeding the training process with fewer parameters than LSTM.

In this study, a GRU with an attention mechanism was employed to forecast future TT (short-term) using past travel time sequences. Fig 1 shows the structure of a GRU cell. The Reset gate decides how much past information the model needs to forget at each timestamp. Likewise, the Update gate is responsible for determining how much past information the model needs to pass at each timestamp. Eqs (1)–(4) show how the two gates govern the flow of information within the GRU network.

$$u_t = \sigma(W^u x_t + U^u h_{t-1}) \tag{1}$$

$$r_t = \sigma(W^r x_t + U^r h_{t-1}) \tag{2}$$

$$h'_t = \mu(W x_t + r_t \odot U h_{t-1}) \tag{3}$$

$$h_t = z_t \odot h_{t-1} + (1 - z_t \odot h'_t) \tag{4}$$

where $r_t$ and $u_t$ denote reset gate and update gate, respectively, $h'_t$, and $h_t$ denote memory content (current) and memory content (final) at time t, respectively, $\sigma$ is the sigmoid activation function and $\mu$ denotes tanh activation function. $W^u$ and $U^u$ are the respective weight matrices of the two gates whereas $\odot$ denotes element-wise multiplication.

The architectural diagram and data flow are illustrated in Figs 2 and 3.

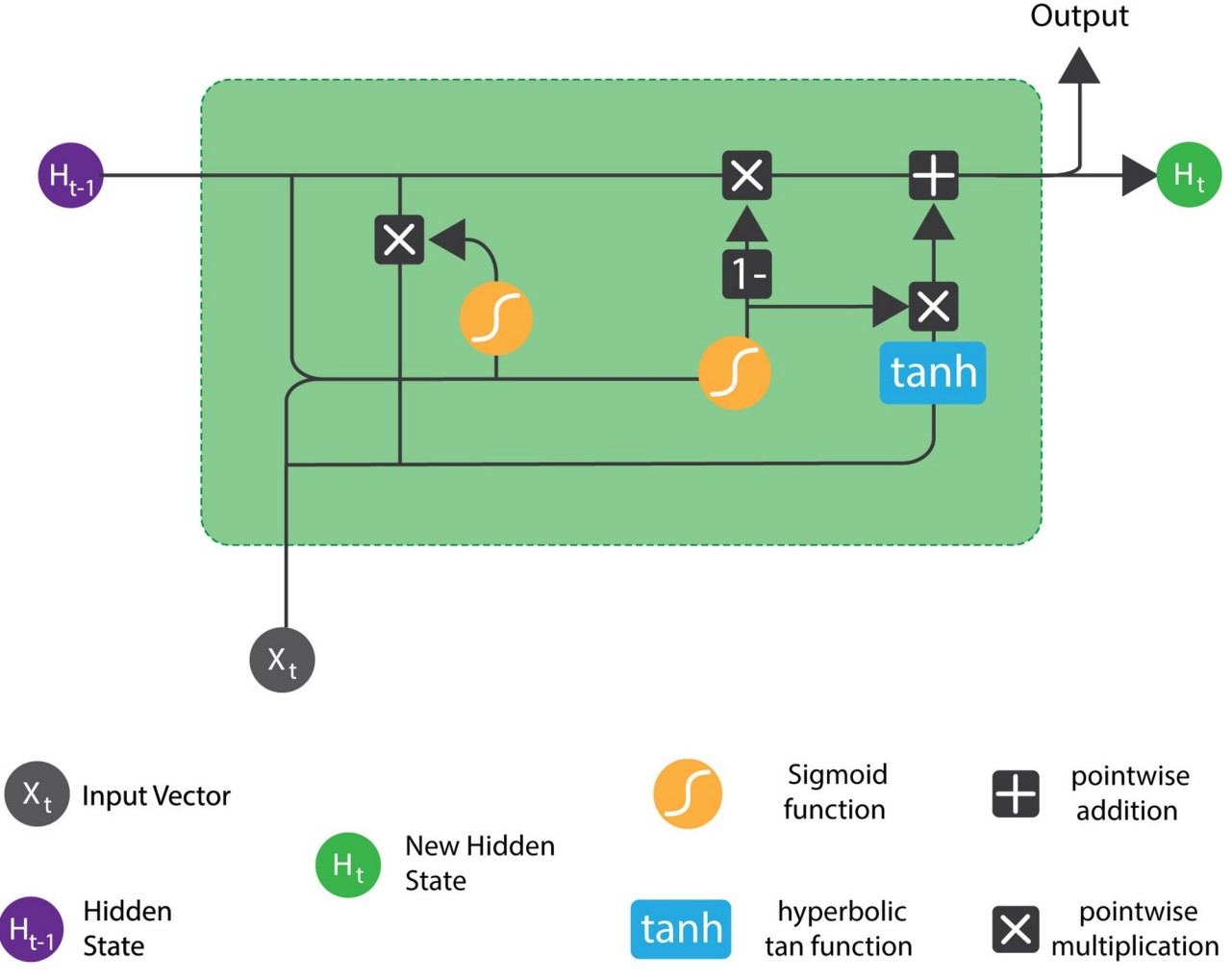

**Fig 1. Structure of a GRU cell** [48].

## Attention mechanism

The attention mechanism enhances the learning ability of predictive models by focusing on relevant information. The authors in [50] improved weight assignment by giving different weights to different text fragments, thereby enhancing the encoding process in neural machine translation. Subsequently, the attention mechanism is successfully applied in document classification [51], image caption generation [52], tabular learning [53], and many more. In the context of ITS, attention has recently been applied to traffic congestion prediction [6], traffic speed prediction, [8], traffic flow prediction [44] and travel time prediction [26]. Because standard RNN models such as LSTM and GRU could not identify the relevance in historical travel time sequences explicitly, we have implemented an attention mechanism to learn global trends in travel time sequences. The following three steps explain the attention mechanism. First, GRU computes the hidden states at different timestamps (H=($h_1$, $h_2$,...,$h_n$)). In the second step, weights of each hidden state $h_i$ are computed using a scoring

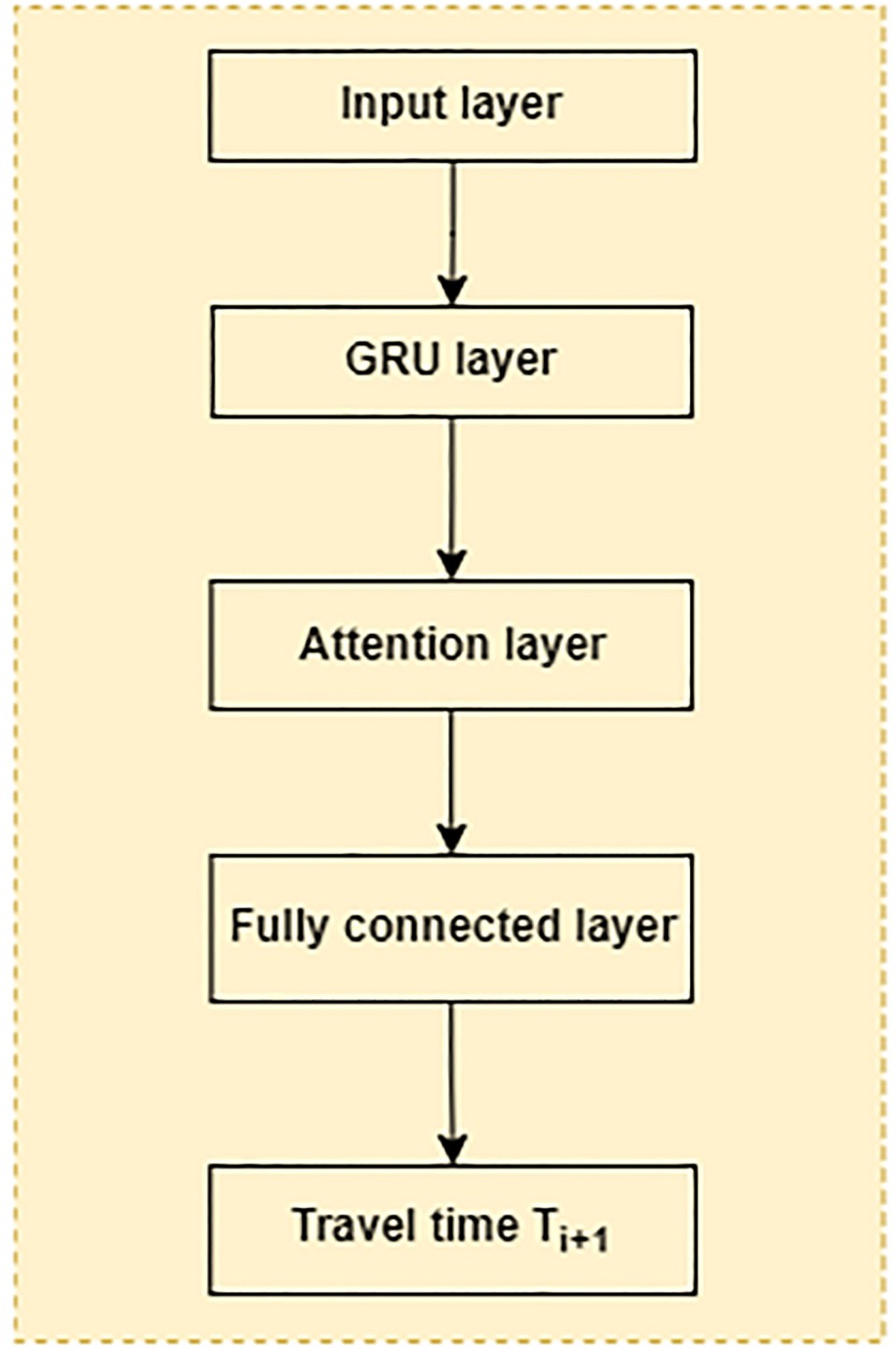

**Fig 2. Layers of the proposed GRU model.**

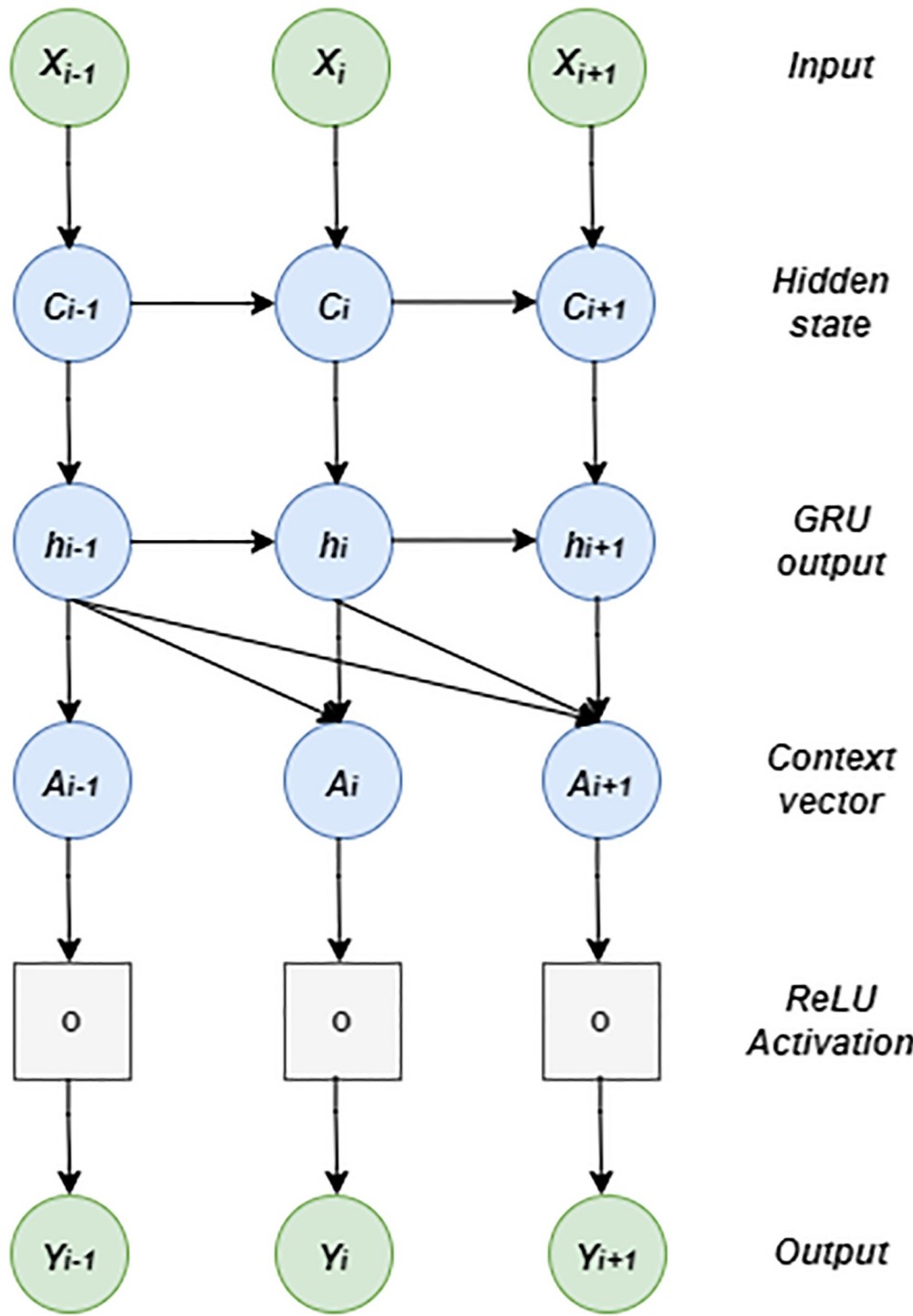

**Fig 3. Data flow of proposed GRU model.**

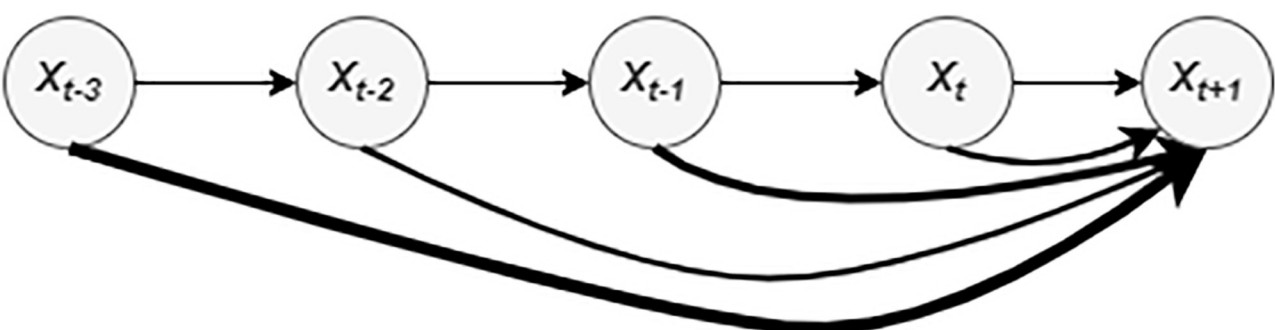

**Fig 4. An illustration of the attention mechanism.**

function (i.e., a two-layer deep neural network in our case). Thirdly, the context vector $A_t$, which is used to get the final prediction, is extracted with an attention function. We illustrated the attention mechanism concept in Fig 4 where the relation between the predicted ($X_{t+1}$) value and historical values ($X_{t-3}$, $X_{t-2}$, $X_{t-1}$, $X_t$) is shown by the thickness of arrows. We adapted the attention mechanism implemented for traffic speed prediction in [54] and given by the Eqs (5)–(7).

$$a_i = W_{(h_2)}(W_{(h_1)}H + b_{(h_1)}) + b_{(h_2)} \tag{5}$$

$$\alpha_i = \frac{exp(a_i)}{\sum_{k=1}^{n} exp(a_k)} \tag{6}$$

$$A_t = \sum_{i=1}^{n} \alpha_i * h_i \tag{7}$$

where the weights and biases of two hidden layers are denoted by $W_{(h1)}$, $W_{(h2)}$, $b_{(h1)}$ and $b_{(h2)}$), respectively, $\alpha_i$ shows the dependency between $h_t$ (i.e., current position at t) and $h_t'$ (i.e., the previous position at $t'$) in H.

## Results

### Dataset

We evaluated our model on the Q-Traffic dataset presented in [55]. The dataset contains 15,073 road segments spanning 738.91 km from April 1, 2017, to May 31, 2017. All the data is collected around the most crowded area of Beijing (i.e., around the 6th ring road). This dataset also incorporates events happening around that time like Summer Palace (May Day), Fish Leong Concert, Chou Chuan-huing Concert, 106th Anniversary of THU and Spring outing, etc., causing a massive increase in traffic congestion than usual traffic. The data is aggregated at a 15-minute time interval on every road. The training-test split is 80-20%. The data is normalized to the interval [0, 1]. Travel time for the next 15 minutes, 30 minutes, 45 minutes, and 60 minutes are predicted in this experiment.

## Performance metrics

We used four evaluation measures to evaluate our proposed model; RMSE, MAE, MAPE, and $R^2$. The RMSE can be computed using Eq (8). These equations are taken from [56].

$$RMSE = \sqrt{\frac{1}{n}\sum_{i=1}^{n}(\widehat{TT_i} - TT_i)^2} \tag{8}$$

where TT_i denotes the actual travel time and $\widehat{TT_i}$ denotes the predicted travel time. MAE is the average absolute error among the TT_i and $\widehat{TT_i}$ and is shown in Eq (9).

$$MAE = \frac{1}{n}\sum_{i=1}^{n}|\widehat{TT_i} - TT_i| \tag{9}$$

MAPE denotes the percentage of the difference between the actual and predicted value and is given in Eq (10). A lower value of MAPE indicates high prediction accuracy.

$$MAPE = \frac{1}{n}\sum_{i=1}^{n}\frac{|\widehat{TT_i} - TT_i|}{TT_i} \tag{10}$$

Eq (11) shows the $R2$, which reflects how much variation the model learns.

$$R^2 = 1 - \frac{\sum_{i=1}^{n}|(\widehat{TT_i} - TT_i)|}{\sum_{i=1}^{n}|(\widehat{TT_i} - TT_m)|} \tag{11}$$

Here TT_m denotes the mean travel time value. For optimal prediction, RMSE and MAE should be zero (or close to zero), and $R2$ should be close to one.

## Hyperparameters setting

In our experiments, we set the training epoch to 600 and the learning rate to 0.001. One of the important hyperparameters is the number of hidden units which greatly affect the prediction output. We tested our model with 8, 16,32,64, and 128 hidden units with varying batch sizes (i.e. [16, 32 and 64] and chose the values with the best results. The results of the finalized model with a batch size of 32 are illustrated in Fig 5. It can be seen that when the hidden units are set to 32, we got the smallest values for RMSE, MAE, and MAPE and higher values for $R^2$. By choosing a smaller value or a value greater than 32 for batch size and hidden units, the evaluation measures either give higher values or start diverging from minima. As a result, we set hidden units to 32 in our experiments. To avoid overfitting, a normalization term is added in loss computation as shown in Eq (12).

$$Loss = |\widehat{TT_i} - TT_i| + CL_{norm} \tag{12}$$

where $L_{norm}$ is the normalization term and C is a parameter whose value is set to 0.0015 for this experiment.

## Baselines

- HA [57]: Historical Average is a simple mathematical model which takes the mean of traffic values in the historical interval as the final prediction.

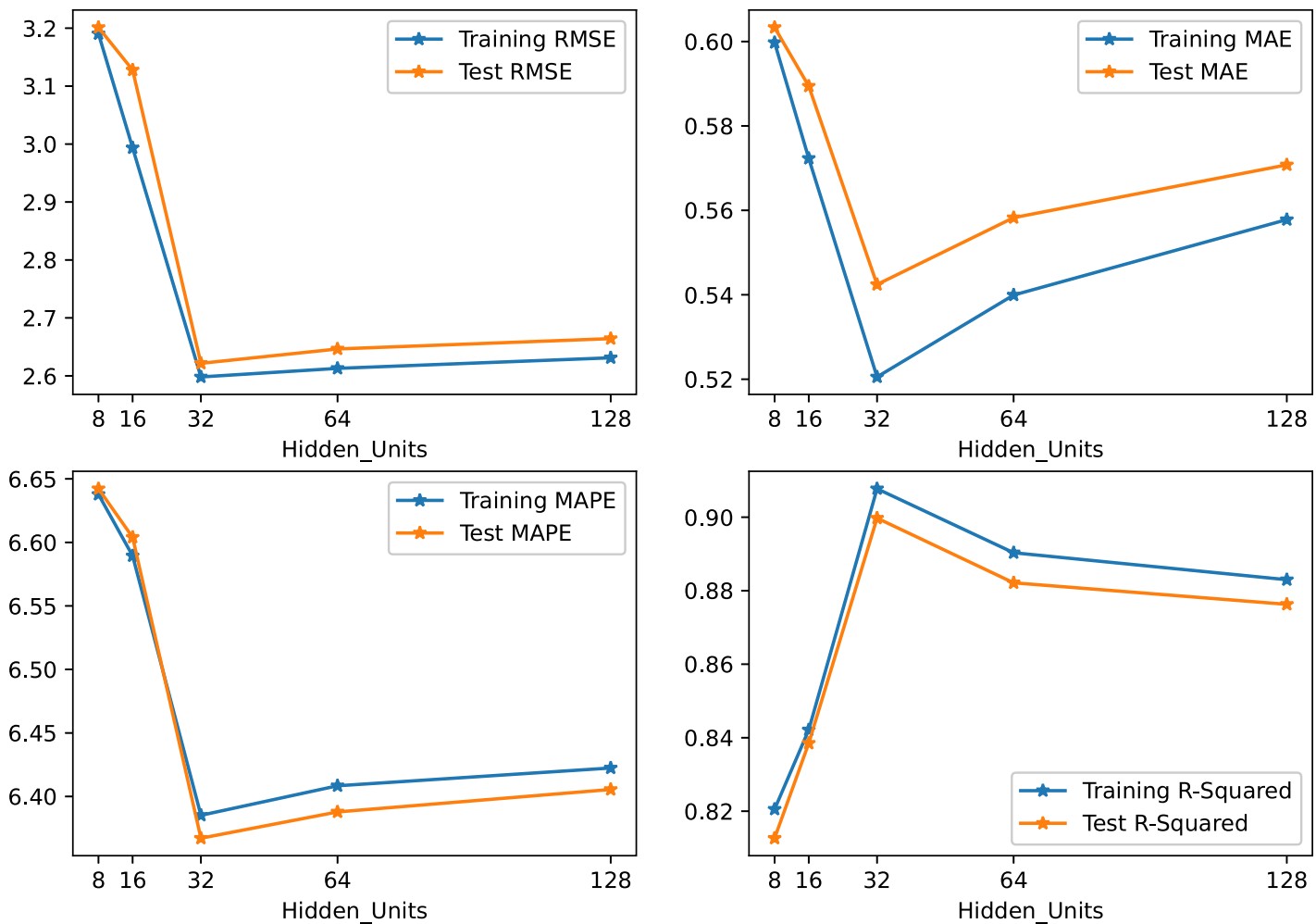

**Fig 5. Comparison of RMSE, MAE, MAPE, and R Squared error results for different hidden units values.**

- ARIMA [58]: ARIMA is a widely used time-series model that predicts future traffic data by fitting a parametric model to the historical time series. We have used ARIMA from *statsmodel* python package with parameters setting as (2,0,1).

- SVR [59]: SVR is a well-known machine learning model that we train on training data to obtain a relationship between explanatory variables and the target variable. In this experiment, we have used *rbf* kernel function, $\epsilon$ =0.1, and C = 1.

- XGBoost [60]: XGBoost is a state-of-the-art model from the decision tree family that employs an ensemble of decision tree regressors for travel time prediction. In our work, we have set *max_depth* to 7.

- MLP [59]: MLP is a feed-forward artificial neural network consisting of fully connected layers (dense). In this study, we used 3 layers deep neural network. Hidden units are set to 64 and *relu* activation function is used.

- GRU [48]: GRU is an improved variant of recurrent neural network (Readers are referred to Section III for details.). In our experiment, a single layer GRU model with 32 hidden units and *relu* activation function is used.

## Performance comparison with baselines

A comparison of the proposed model and baseline approaches for the prediction horizon of 15 minutes, 30 minutes, 45 minutes, and 60 minutes is shown in Table 1. ✳ shows minimal values of the error measures indicating the model's poor performance.

**Table 1. Performance evaluation of baselines & proposed (Overall).**

| Predication-Horizon | Model | RMSE | MAE | MAPE | $R^2$ |
|---|---|---|---|---|---|
| 15-min | HA | 2.73136 | 0.60377 | 6.61174 | 0.87217 |
| | ARIMA | 4.25444 | 1.48836 | 9.24698 | ✳ |
| | SVR | 2.71936 | 0.58616 | 6.58210 | 0.87379 |
| | XGBoost | 2.69701 | 0.58133 | 6.56837 | 0.87706 |
| | MLP | 2.67871 | 0.57979 | 6.54887 | 0.87915 |
| | GRU | 2.63252 | 0.55150 | 6.48860 | 0.88855 |
| | Proposed | **2.62162** | **0.54244** | **6.36716** | **0.89974** |
| | Improvement | **0.41%** | **1.64%** | **1.87%** | **1.26%** |
| 30-min | HA | 2.73136 | 0.60377 | 6.61174 | 0.87217 |
| | ARIMA | 4.05522 | 1.29382 | 9.49330 | ✳ |
| | SVR | 2.73877 | 0.59465 | 6.65536 | 0.87095 |
| | XGBoost | 2.70256 | 0.58505 | 6.59215 | 0.87494 |
| | MLP | 2.69742 | 0.58291 | 6.56015 | 0.87514 |
| | GRU | 2.65508 | 0.57152 | 6.50139 | 0.88298 |
| | Proposed | **2.64053** | **0.56109** | **6.38909** | **0.89177** |
| | Improvement | **0.55%** | **1.82%** | **1.73%** | **0.99%** |
| 45-min | HA | 2.73136 | 0.60377 | 6.61174 | 0.87217 |
| | ARIMA | 4.02042 | 1.27711 | 9.51899 | ✳ |
| | SVR | 2.75216 | 0.61031 | 6.68810 | 0.86794 |
| | XGBoost | 2.72788 | 0.60527 | 6.61623 | 0.87028 |
| | MLP | 2.70909 | 0.59954 | 6.58186 | 0.87136 |
| | GRU | 2.67372 | 0.59137 | 6.52952 | 0.87966 |
| | Proposed | **2.66211** | **0.57983** | **6.40188** | **0.88548** |
| | Improvement | **0.43%** | **1.95%** | **1.95%** | **0.66%** |
| 60-min | HA | 2.73136 | 0.60377 | 6.61174 | 0.87217 |
| | ARIMA | 3.99637 | 1.26543 | 9.52668 | ✳ |
| | SVR | 2.77696 | 0.62892 | 6.70206 | 0.86491 |
| | XGBoost | 2.74182 | 0.62154 | 6.64527 | 0.86725 |
| | MLP | 2.73427 | 0.61899 | 6.60636 | 0.86842 |
| | GRU | 2.69637 | 0.61543 | 6.54968 | 0.87395 |
| | Proposed | **2.68005** | **0.59523** | **6.42337** | **0.87893** |
| | Improvement | **0.61%** | **3.28%** | **1.93%** | **0.57%** |
| Overall | HA | 2.73136 | 0.60377 | 6.61174 | 0.87217 |
| | ARIMA | 4.08161 | 1.33118 | 9.44649 | ✳ |
| | SVR | 2.74681 | 0.0501 | 6.65691 | 0.86940 |
| | XGBoost | 2.71732 | 0.59830 | 6.64527 | 0.87238 |
| | MLP | 2.70237 | 0.59531 | 6.57431 | 0.87352 |
| | GRU | 2.66442 | 0.58246 | 6.51730 | 0.88129 |
| | Proposed | **2.65108** | **0.56965** | **6.39538** | **0.88898** |
| | Improvement | **0.50%** | **2.20%** | **1.87%** | **0.87%** |

The results demonstrate that SVR with a non-linear kernel performed better than HA and ARIMA. For example, SVR reduces the RMSE from 2.73136 (HA) and 4.25444 (ARIMA) to 2.71936, a 0.44 percent and 36.08 percent reduction, respectively. Compared to HA, ARIMA, and SVR, an ensemble model XGBoost performed better. For example, XGBoost shows a reduction of 1.26%, 36.61%, and 0.82% against HA, ARIMA, and SVR in RMSE for the prediction horizon of 15 minutes. Similarly, there is a reduction of 3.72%, 60.94%, and 0.82% in MAE when comparing XGBoost with HA, ARIMA, and SVR for the same prediction horizon. The same trend is visible for MAPE in Table 1. Like RMSE, MAE, and MAPE, we observed an improvement in $R^2$ error compared to HA, ARIMA, and SVR. For instance, an improvement of 0.56% is reported when comparing HA with XGBoost.

The results in the Table 1 show that neural networks such as MLP, GRU, and our proposed attention-based GRU model performed better than traditional machine learning and time-series models. For example, there is a reduction of 1.93%, 37.04%, 1.49%, and 0.68% in RMSE when comparing MLP with HA, ARIMA, SVR, and XGBoost for the prediction horizon of 15 minutes. Likewise, compared to HA, ARIMA, SVR, XGBoost, and MLP, GRU performance was reduced by 3.62%, 38.12%, 3.19%, 2.39%, and 1.72%, in terms of RMSE. Our proposed attention-based GRU has shown a decrease in RMSE, MAE, and MAPE of 4.02%, 10.16%, and 3.7%, compared to HA. Furthermore, comparing HA with our proposed model, we have seen an improvement of about 3.16% in the $R^2$ error. With the attention mechanism, we improved the prediction precision of GRU. The RMSE, MAE, and MAPE have been decreased by 0.41 percent, 1.64 percent, and 1.87 percent, respectively.

Our proposed model can achieve better prediction performance regardless of how the horizon varies, and the prediction results have a lower tendency to change. Compared to GRU, there is a 0.50 percent, 2.20 percent, and 1.87 percent reduction in RMSE, MAE, and MAPE, respectively, demonstrating that the proposed model can be utilized for both short and long-term prediction without reducing performance significantly.

To demonstrate our model performance, we selected a single road and showed the plots of actual and predicted travel time values for the four prediction horizons. The visualization results on test data for the prediction horizon of 15-min, 30-min, 45-min, and 60-min are shown in Figs 6–13. These results show that our proposed model captures the traffic dynamics

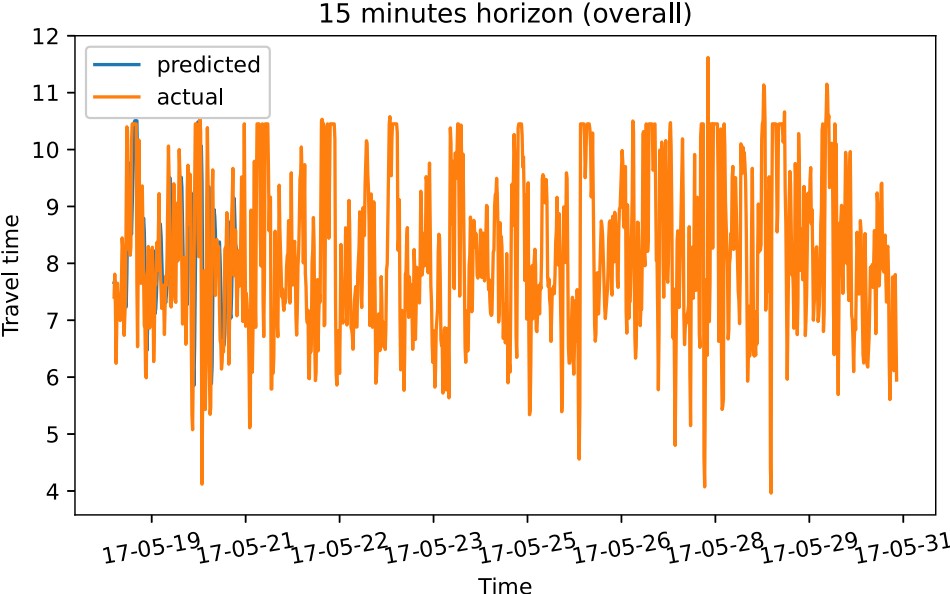

**Fig 6. Prediction results for 15 minutes horizon on test data (overall).**

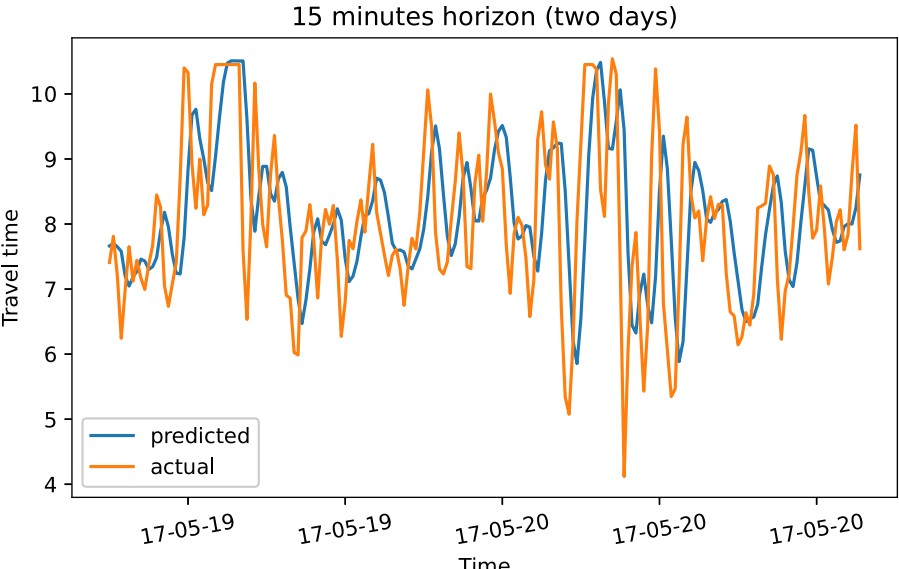

**Fig 7. Prediction results for 15 minutes horizon on test data (two days).**

regardless of the prediction horizon. However, taking into account both the geographical and temporal dimensions can improve the results even more, particularly along with local minima/maxima.

## Robustness analysis

Noise is unavoidable during the data collection process in real-world circumstances. We have performed a robustness analysis to test the generalization of our proposed model in the presence of noise.

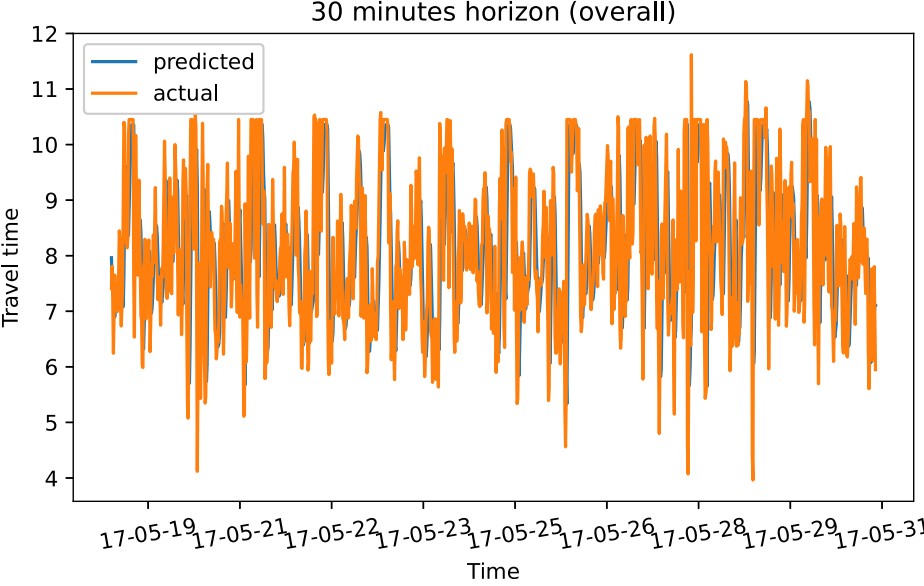

**Fig 8. Prediction results for 30 minutes horizon on test data (overall).**

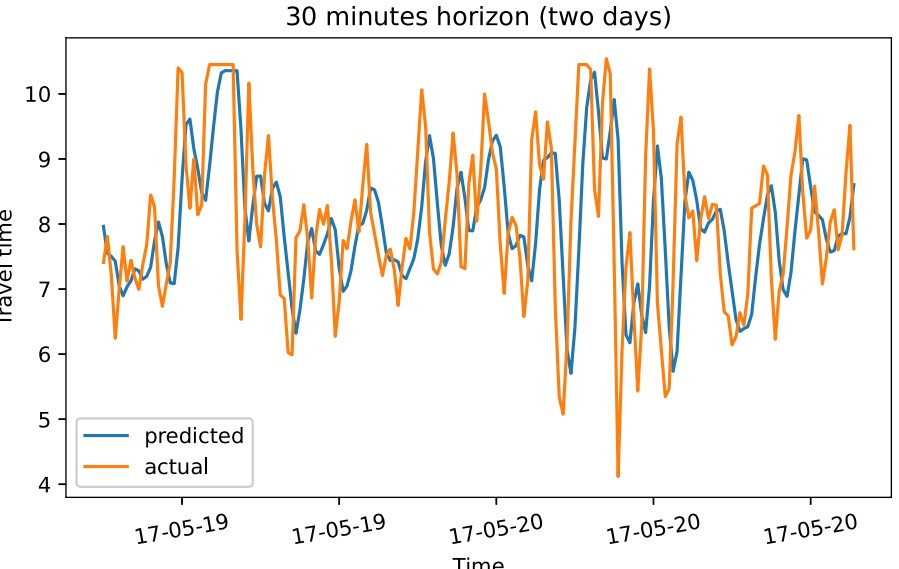

**Fig 9. Prediction results for 30 minutes horizon on test data (two days).**

We induced a common type of noise that obeys Gaussian Distribution (i.e., $N \in 0, \sigma^2$ where *sigma* varies from 0.2−2) to our dataset after normalising it to the interval [0, 1]. Fig 14 shows that the change in error measures is small, which shows the proposed model's generalization ability to perform well even on noisy traffic data.

## Conclusion

Travel time is becoming an attractive research area in the traffic prediction domain compared to other traffic variables as it is more interpretable and understandable for those unfamiliar with transportation terms. With cutting-edge traffic data collection technologies in recent

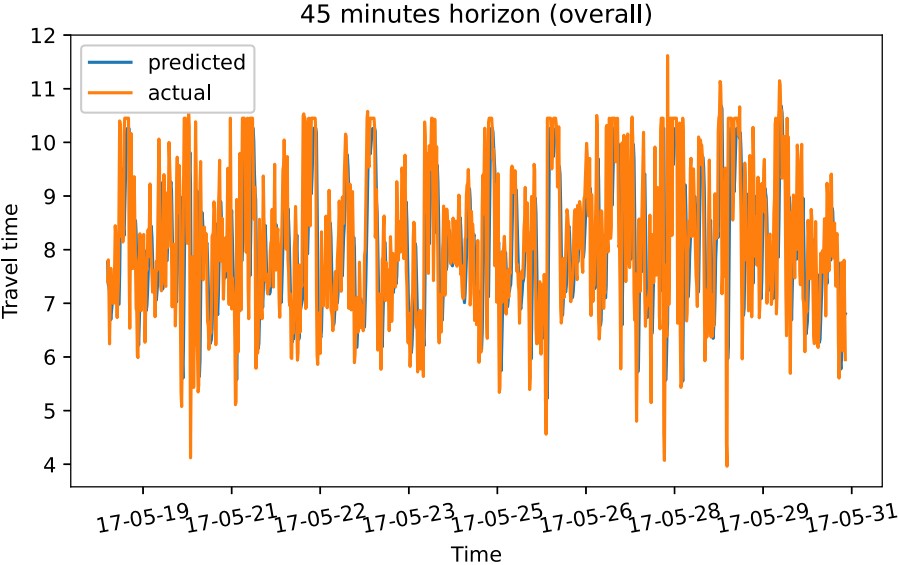

**Fig 10. Prediction results for 45 minutes horizon on test data (overall).**

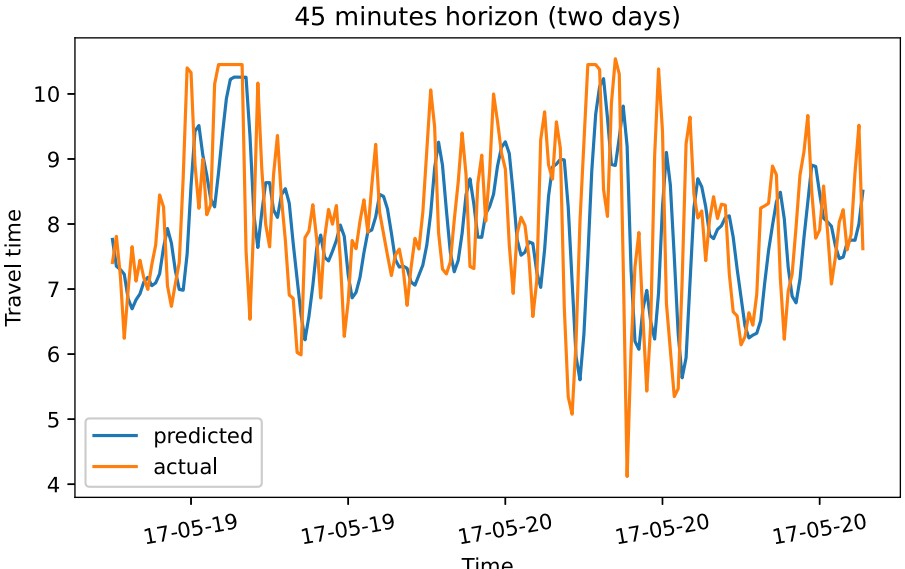

**Fig 11. Prediction results for 45 minutes horizon on test data (two days).**

years, data-driven approaches have been extensively applied for traffic prediction problems. In this article, we have implemented a GRU model to capture temporal relations in travel time data. We added an attention mechanism to the GRU model to help it learn the relevant information and enhance prediction precision. Experimental results show improvement in the performance when using the proposed attention-based GRU model compared to classical time-series models. Furthermore, we conducted a robustness test and found that the proposed model performed better even when there was noise in the traffic data, which is inevitable, with just a minor degradation in the model's performance.

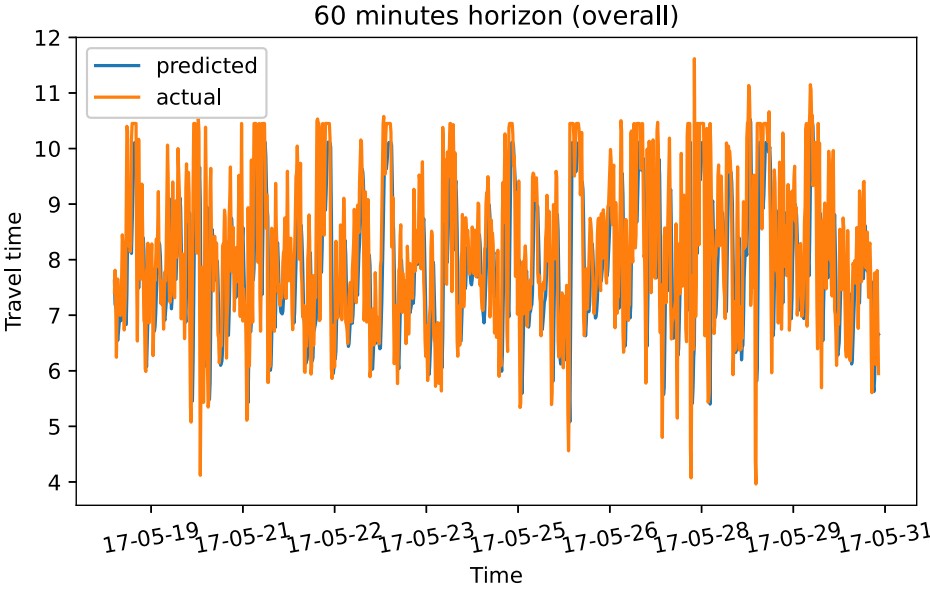

**Fig 12. Prediction results for 60 minutes horizon on test data (overall).**

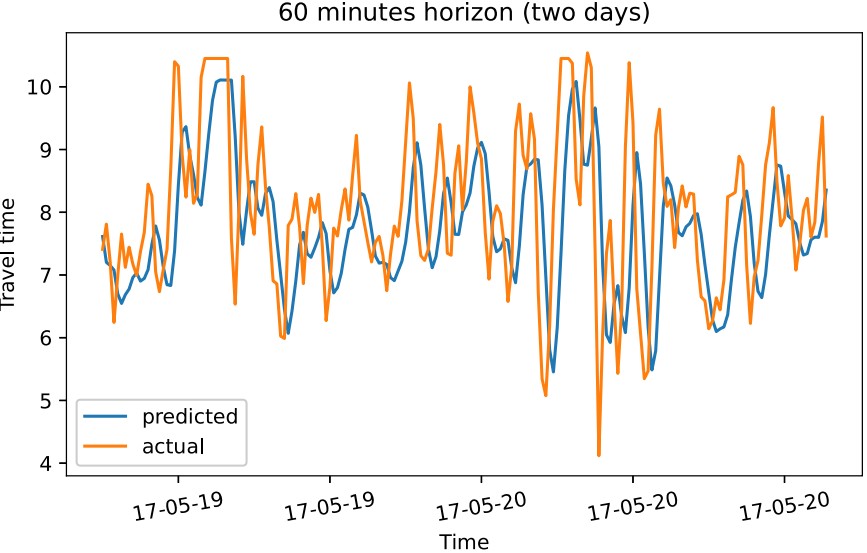

**Fig 13. Prediction results for 60 minutes horizon on test data (two days).**

In the future, we plan to extend our work by incorporating graph-based neural networks to cater to the spatial dimension along with temporal on the same dataset To improve prediction accuracy, we also plan to combine exogenous elements such as weather, peak/non-peak hours, and other factors with traffic data.

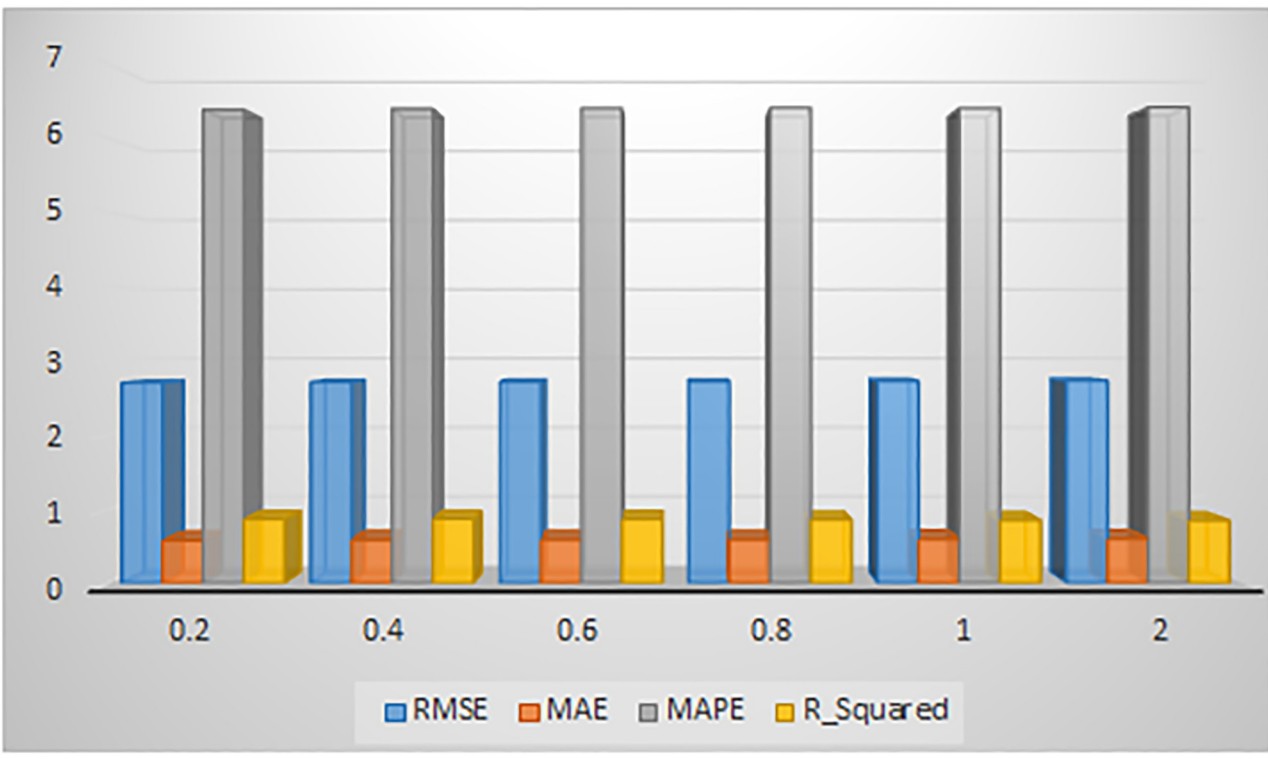

**Fig 14. Robustness analysis after adding Gaussian noise.**

## Author Contributions

**Conceptualization:** Jawad-ur-Rehman Chughtai.

**Data curation:** Jawad-ur-Rehman Chughtai.

**Funding acquisition:** Muhammad Muneeb.

**Methodology:** Jawad-ur-Rehman Chughtai.

**Supervision:** Irfan Ul Haq.

**Validation:** Jawad-ur-Rehman Chughtai.

**Visualization:** Jawad-ur-Rehman Chughtai.

**Writing – original draft:** Jawad-ur-Rehman Chughtai.

**Writing – review & editing:** Jawad-ur-Rehman Chughtai, Irfan Ul Haq, Muhammad Muneeb.

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
