## [Decision Letter · Decision Letter 0]

29 Aug 2022

PONE-D-22-18546An Attention Based Recurrent Learning Model for Short-term Travel Time PredictionPLOS ONE

Dear Dr. Chughtai,

Thank you for submitting your manuscript to PLOS ONE. After careful consideration, we feel that it has merit but does not fully meet PLOS ONE’s publication criteria as it currently stands. Therefore, we invite you to submit a revised version of the manuscript that addresses the points raised during the review process.

We look forward to receiving your revised manuscript.

Kind regards,

Xiyu Liu

Academic Editor

PLOS ONE

Journal Requirements:

“The funding for this research is provided by the Khalifa University of Science and Technology.”

Reviewers' comments:

Reviewer's Responses to Questions

**Comments to the Author**

1. Is the manuscript technically sound, and do the data support the conclusions?

Reviewer #1: Yes

Reviewer #2: Yes

Reviewer #3: Yes

2. Has the statistical analysis been performed appropriately and rigorously? 

Reviewer #1: Yes

Reviewer #2: Yes

Reviewer #3: Yes

3. Have the authors made all data underlying the findings in their manuscript fully available?

Reviewer #1: Yes

Reviewer #2: Yes

Reviewer #3: Yes

4. Is the manuscript presented in an intelligible fashion and written in standard English?

Reviewer #1: Yes

Reviewer #2: Yes

Reviewer #3: Yes

5. Review Comments to the Author

Reviewer #1: Thank you very much for the beautiful manuscript. I think the paper is in good condition!

The idea presented in the attention section is very practical and logical.

Manuscripts and forms are written in beautiful formats.

Accept

Reviewer #2: Paper summary:

This paper proposed an attention-based GRU model for short-term travel time prediction, enabling GRU to learn the relevant context in historical time slots and update the weights of hidden states accordingly. The authors evaluated the proposed model using FCD data from Beijing. To demonstrate the generalization of the proposed model, the authors performed a robustness analysis by adding noise obeying Gaussian distribution. The experimental results on test data indicated that the proposed model performed better than the existing deep learning time-series models in terms of RMSE, MAE, MAPE, and R^2.

Strengths:

1. This proposal is the first to use traffic flow as input with attention-based GRU to forecast travel time.

2. This proposal is well-structured, and the experiments are detailed and convincing.

Weaknesses:

1. I am wondering whether the overfitting of the original model will affect the results of this proposal.

2. The related work section is suggested to be divided into 2-3 subsections to make the structure of this section clearer.

Other comments:

1. It seems that attention based is missing a hyphen in “attention based”.

2. To overcome these limitations, two specialized variants of RNNs, Gated Recurrent Unit (GRU) [48] and Long Short-Term Memory (LSTM) [49] are developed. Consider inserting a comma to separate the elements.

3. RNNs, unlike MLPs and CNNs (feed-forward neural networks and take data all at once), act on data sequentially and are frequently employed in the Natural Language Processing (NLP) domain. It seems that there is a grammar mistake.

4. “This paper employed GRU with attention mechanism to process travel time sequences and forecast future TT.”. It seems that there is an article usage problem.

Reviewer #3: The authors proposed an attention-based GRU model for short-term travel time prediction to cope with this problem enabling GRU to learn the relevant context in historical time slots and update the weights of hidden states accordingly. This scheme thus solves the problem that the existing GRU does not consider the relationship between various historical travel time slots for traffic prediction. The main message, background and figures are generally clear and concise, but some comparisons with the state-of-the-art algorithms still need to be added in the experimental results section. I recommend improving the following points.

==Major concern==

1.The algorithm proposed in the article integrates the attention mechanism into the GRU to improve the prediction ability. And the comparison results with GRU and several other traditional algorithms are given in the article. In fact, in recent years, attention mechanism has been introduced into DNN to solve Travel Time Prediction, such as [1]. Therefore, it is hoped that the author can add new experiments to compare with such algorithms.

[1] Wu J, Wu Q, Shen J, Cai C. Towards attention-based convolutional long short-term memory for travel time prediction of bus journeys. Sensors. 2020;20(12):3354

==Minor concern==

1. On line 203 of the article, too many commas are printed.

2. References are poorly written. For example, the representation of page numbers, some are p.785-794, some are: 785-794. In addition, the reference is ended with a period, and some ends with a semicolon and a period. and many more.

6. PLOS authors have the option to publish the peer review history of their article (what does this mean?). If published, this will include your full peer review and any attached files.

Reviewer #1: No

Reviewer #2: No

Reviewer #3: No

---

## [Author Response · Author response to Decision Letter 0]

21 Oct 2022

1. Reviewer#2, Concern # 1: I am wondering whether the overfitting of the original model will affect the results of this proposal.

Author response: Thank you very much for the valuable suggestion.

Author action: As discussed in the Dataset section, we have used holdout cross validation (80-20 %) to validate the results of our proposed approach. Furthermore, we have used a normalization term in the loss function calculation to regularize the training and improve model generalization. 

2. Reviewer#2, Concern # 2: The related work section is suggested to be divided into 2-3 subsections to make the structure of this section clearer. 

Author response: Thank you very much for the valuable suggestion.

Author action: We have updated the Related Work section by organizing the literature under suitable headings as suggested by the reviewer. 

3. Reviewer#2, Concern # 3: It seems that attention based is missing a hyphen in “attention based”.

Author response: Thank you very much for pointing it out.

Author action: We have updated the manuscript and added a hyphen in “attention based”.

4. Reviewer#2, Concern # 4: To overcome these limitations, two specialized variants of RNNs, Gated Recurrent Unit (GRU) [48] and Long Short-Term Memory (LSTM) [49] are developed. Consider inserting a comma to separate the elements.

Author response: Thank you very much for the valuable suggestion.

Author action: We have inserted comma to separate the elements as suggested by the reviewer.

5. Reviewer#2, Concern # 5: RNNs, unlike MLPs and CNNs (feed-forward neural networks that take data all at once), act on data sequentially and are frequently employed in the Natural Language Processing (NLP) domain. It seems that there is a grammar mistake.

Author response: Thank you very much for pointing it out.

Author action: We have corrected the identified grammar mistake and updated the manuscript. 

6. Reviewer#2, Concern # 6: “This paper employed GRU with attention mechanism to process travel time sequences and forecast future TT.”. It seems that there is an article usage problem.

Author response: Thank you very much for pointing it out.

Author action: We have corrected the highlighted issue in the manuscript.

1. Reviewer#3, Concern # 1: The algorithm proposed in the article integrates the attention mechanism into the GRU to improve the prediction ability. And the comparison results with GRU and several other traditional algorithms are given in the article. In fact, in recent years, attention mechanism has been introduced into DNN to solve Travel Time Prediction, such as [1]. Therefore, it is hoped that the author can add new experiments to compare with such algorithms.

[1] Wu J, Wu Q, Shen J, Cai C. Towards attention-based convolutional long short-term memory for travel time prediction of bus journeys. Sensors. 2020;20(12):3354

Author response: Thank you very much for the valuable suggestion.

Author action: We have cited the reference suggested by the reviewer in our paper and some other references in the Related Work section. This research work is different from the predictions provided by [1] and [2]. [1] proposed an attention-based convolutional long short-term memory for predicting journey trip time of selected bus routes at current time. [2] proposed attention-based LSTM for predicting travel time of freeway at current time. In our research work, we have used urban network data of Beijing which is different from freeways or bus data (selected routes). Secondly, we proposed an attention-based GRU model for short-term travel time prediction. This is the reason we have compared our approach with GRU and several other traditional algorithms from various families as discussed in the Related Work section. A fair comparison of real-time (current time) prediction approaches with short-term travel time prediction approaches is not possible. Once again, we are thankful to the respectable reviewer for the valuable suggestion.

[1] Wu J, Wu Q, Shen J, Cai C. Towards attention-based convolutional long short-term memory for travel time prediction of bus journeys. Sensors. 2020;20(12):3354

[2] Ran X, Shan Z, Fang Y, Lin C. An LSTM-based method with attention

mechanism for travel time prediction. Sensors. 2019;19(4):861.

2. Reviewer#3, Concern # 2: On line 203 of the article, too many commas are printed.

Author response: Thank you very much for pointing it out.

Author action: We have corrected the identified error in the manuscript.

3. Reviewer#3, Concern # 3: References are poorly written. For example, the representation of page numbers, some are p.785-794, some are: 785-794. In addition, the reference is ended with a period, and some ends with a semicolon and a period. and many more.

Author response: Thank you very much for pointing it out. 

Author action: We have updated the manuscript by updating all the references.

---

## [Decision Letter · Decision Letter 1]

9 Nov 2022

An Attention-based Recurrent Learning Model for Short-term Travel Time Prediction

PONE-D-22-18546R1

Dear Dr. Chughtai,

We’re pleased to inform you that your manuscript has been judged scientifically suitable for publication and will be formally accepted for publication once it meets all outstanding technical requirements.

Kind regards,

Xiyu Liu

Academic Editor

PLOS ONE

Reviewers' comments:

Reviewer's Responses to Questions

**Comments to the Author**

1. If the authors have adequately addressed your comments raised in a previous round of review and you feel that this manuscript is now acceptable for publication, you may indicate that here to bypass the “Comments to the Author” section, enter your conflict of interest statement in the “Confidential to Editor” section, and submit your "Accept" recommendation.

Reviewer #2: (No Response)

Reviewer #3: All comments have been addressed

2. Is the manuscript technically sound, and do the data support the conclusions?

Reviewer #2: (No Response)

Reviewer #3: Yes

3. Has the statistical analysis been performed appropriately and rigorously? 

Reviewer #2: (No Response)

Reviewer #3: Yes

4. Have the authors made all data underlying the findings in their manuscript fully available?

Reviewer #2: (No Response)

Reviewer #3: Yes

5. Is the manuscript presented in an intelligible fashion and written in standard English?

Reviewer #2: (No Response)

Reviewer #3: Yes

6. Review Comments to the Author

Reviewer #2: (No Response)

Reviewer #3: The authors proposed an attention-based GRU model for short-term travel time prediction to cope with this problem enabling GRU to learn the relevant context in historical time slots and update the weights of hidden states accordingly.They have answered all my questions. I have no other comments.

7. PLOS authors have the option to publish the peer review history of their article (what does this mean?). If published, this will include your full peer review and any attached files.

Reviewer #2: No

Reviewer #3: No

---

## [Editor Report · Acceptance letter]

18 Nov 2022

PONE-D-22-18546R1 

An Attention-based Recurrent Learning Model for Short-term
Travel Time Prediction 

Dear Dr. Chughtai:

I'm pleased to inform you that your manuscript has been deemed suitable for publication in PLOS ONE. Congratulations! Your manuscript is now with our production department. 

Kind regards, 

on behalf of

Professor Xiyu Liu 

Academic Editor

PLOS ONE